# The “Selfie Test”: A Novel Test for the Diagnosis of Lateral Epicondylitis

**DOI:** 10.3390/medicina59061159

**Published:** 2023-06-16

**Authors:** Shai Factor, Pablo Gabriel Snopik, Assaf Albagli, Ehud Rath, Eyal Amar, Franck Atlan, Guy Morag

**Affiliations:** 1Department of Orthopedic Surgery, Tel Aviv Medical Center, 6 Weitzman St., Tel Aviv 6423906, Israel; 2Sackler Faculty of Medicine, Tel Aviv University, Tel Aviv 6423906, Israel; 3Clalit Health Services, Beer Sheva 8489428, Israel

**Keywords:** lateral epicondylitis, tennis elbow, provocative tests, diagnostic accuracy, selfie test

## Abstract

*Background*: Lateral epicondylitis (LE) is one of the most diagnosed elbow pathologies. The purpose of this study was to determine the diagnostic test accuracy of a new test (selfie test) for the diagnosis of LE. *Methods*: Medical data were collected from adult patients who presented with LE symptoms and ultrasound findings that supported the diagnosis. Patients underwent a physical examination, including provocative tests for diagnosis as well as the selfie test, and were asked to fill out the Patient-Rated Tennis Elbow Evaluation (PRTEE) questionnaire and subjectively rate the activity of their affected elbow. *Results*: Thirty patients were included in this study (seventeen females, 57%). The mean age was 50.1 years old (range of 35 to 68 years). The average duration of symptoms was 7 ± 3.1 months (range of 2 to 14 months). The mean PRTEE score was 61.5 ± 16.1 (range of 35 to 98), and the mean subjective elbow score was 63 ± 14.2 (range of 30 to 80). Mill’s, Maudsley’s, Cozen’s, and the selfie tests had sensitivities of 0.867, 0.833, 0.967, and 0.933, respectively, with corresponding positive predictive values of 0.867, 0.833, 0.967, and 0.933. *Conclusions*: The selfie test’s active nature, which allows patients to perform the assessment themselves, could be a valuable addition to the diagnostic process, potentially improving the accuracy of the diagnosis of LE (levels of evidence: IV).

## 1. Introduction

Lateral epicondylitis (LE), also known as tennis elbow, is one of the most diagnosed elbow pathologies, with a prevalence of 1–2% of the general population [1]. LE is associated with a history of repetitive eccentric movements of wrist extension, leading to an overload on the origin of the extensor carpi radialis brevis (ECRB) tendon and resulting in microtears [2,3]. 

The pathophysiology of lateral epicondylitis is marked by angio-fibroblastic dysplasia, which is associated with the formation of granulation tissue, microtears, vascular hyperplasia, and disorganized collagen. It should be noted that inflammation is typically not present in lateral epicondylitis, although it may occur in the early stages of the disease [4].

Patients commonly report experiencing pain at or around the lateral epicondyle, which may radiate down the forearm in line with the common extensor muscle mass, and sometimes upward into the upper arm. This pain is typically triggered or aggravated by the contraction of the common extensor mass during various activities. The pain’s intensity can vary, ranging from occasional and mild to continuous and severe, impacting daily activities and even causing sleep disturbance at night [5]. Rest, modifying activities, and conservative treatments are typically effective in alleviating symptoms for most patients [6].

The diagnosis of LE is mainly clinical. A detailed evaluation, including history and a physical examination, is of paramount importance. The hallmark of the physical examination is local tenderness over the origin of the ECRB at the lateral epicondyle, and symptoms are often reproduced through provocative tests [2]. These tests are based on the passive stretching of the common extensor or the active recruitment of these muscles. The most widely accepted and used tests are Cozen’s, Mill’s, and Maudsley’s [7]. Although limited literature is available on the accuracy of these tests for diagnosing LE, a recent systematic review found that Cozen’s test had the highest sensitivity at 91%, followed by Mill’s test at 76% and Maudsley’s test at 70%; however, the specificity of these tests is generally low [8].

As the diagnosis is clinically based, imaging is often unnecessary, although it can assist in more complex cases [9]. Ultrasonography (US) has been proven to have high sensitivity but low specificity in the diagnosis of LE, and is considered the most useful tool; however, it suggested that it should not be used in isolation, but rather as a complementary component of the overall assessment [8,10,11].

With technology advancing at a rapid pace, repetitive actions are no longer limited to sports and manual labor, but also include the extensive use of cellular phones, which have become a part of our daily routine [12]. Additionally, the advancement of telemedicine and the need for rapid diagnoses without a physical examination by a physician are gaining momentum and becoming a significant aspect of the diagnostic process. As such, it is essential to develop new tools and tests that are adapted to our changing daily activities [13,14]. 

Furthermore, the integration of smartphones and other mobile devices into various aspects of healthcare has revolutionized the way that we access medical information and resources. Mobile applications and wearable devices now allow individuals to monitor their health, track symptoms, and even receive personalized treatment plans. This shift towards mobile healthcare empowers individuals to take a proactive role in managing their well-being, enabling early interventions and preventing the exacerbation of health conditions [13].

The purpose of this study was to evaluate the accuracy of the selfie test, a new diagnostic tool for LE. The test is performed entirely by a patient and assesses pain in the lateral aspect of the elbow while simulating taking a selfie with a cell phone. We hypothesized that the selfie test would demonstrate an accuracy rate similar to that of other tests.

## 2. Materials and Methods

### 2.1. Study Population

After receiving the approval of the Helsinki Committee, medical data were collected from the medical charts of adult patients who sought examination at the outpatient clinics of the hospital due to elbow pain. The inclusion criteria were patients over 18 years of age who presented with recurrent or new persistent lateral elbow pain and ultrasound findings that supported the diagnosis of lateral epicondylitis. These findings included thickening of the common extensor tendon at its origin, hypoechoic areas within the tendon indicating degeneration or tears, and the irregularity of the cortex (i.e., bone spurs) of the lateral epicondyle. The exclusion criteria included being under 18 years of age, a diagnosis of cervical radiculopathy or myelopathy, having neurological disorders, and having associated elbow fractures (Table 1).

All US evaluations were conducted by an expert US technician who was blinded to the study purpose, and the US assessments were performed prior to any clinical evaluation. Additionally, any US obtained externally were excluded from the analysis.

### 2.2. Data Collection

The data included the patients’ age, sex, dominant hand, affected hand, and duration of symptoms. All patients underwent a physical examination, which included provocative tests for diagnosing LE and the selfie test. Patients were asked to fill out a dedicated Patient-Rated Tennis Elbow Evaluation (PRTEE) questionnaire for tennis elbow [15]. The PRTEE questionnaire is a self-administered tool that comprises 15 questions related to pain and function during the preceding week. It generates two scores: a pain score and a functional score. Each score ranges from 0 to 100, with higher scores indicating more severe symptoms. Additionally, patients were asked to subjectively quantify the level of activity of their affected elbow compared to the unaffected one on a scale of 0 to 100, with 0 representing extreme pain and no function and 100 representing a normal elbow. This additional information helped to provide a more comprehensive assessment of a patient’s condition and provided a better understanding of the impact of their symptoms on daily activities.

### 2.3. Provocative Tests

Mill’s Test

The patient was seated, with the shoulder minimally abducted and the elbow flexed at 90°. The examiner then palpated the patient’s lateral epicondyle with one hand while fully flexing the wrist and pronating the forearm, followed by the extension of the elbow. Pain production on the lateral epicondyle region indicated a positive test [16].

Cozen’s Test

The patient was seated, with their elbow in full extension and forearm in maximal pronation. The patient was asked to make a fist and radially deviate his hand, then to perform wrist extension against resistance. The test was considered positive if pain over the lateral epicondyle was elicited [8].

Maudsley’s Test

The patient was seated, with their elbow flexed to 90° and forearm pronated. The examiner resists the extension of the 3rd finger while palpating the patient’s lateral epicondyle. Pain over the lateral aspect elbow joint indicated a positive test [17].

Selfie Test

The patient was either seated or standing, holding a cell phone with their elbow fully extended. The wrist was then flexed, and the patient was asked to press their thumb independently on the phone screen or top button (simulating the maneuver of taking a selfie picture). Pain over the lateral aspect of the elbow joint indicated a positive test (Figure 1).

### 2.4. Statistical Analysis

Descriptive statistics are used to present quantitative descriptions of data. Since no comparison was made between variables, the use of inferential statistics was obviated. Data are presented in their raw form. Demographic data also presented as mean and standard deviation (SD). In this study, it is important to clarify that a formal analysis to determine the sample size was not conducted. Instead, an estimated sample size was derived based on the recruitment numbers from two previous studies which had enrolled a similar number of patients. Given the specific objectives of the current study, it was determined that a sample size of 30 patients would be sufficient. Furthermore, certain criteria were applied to exclude patients from the study. Specifically, individuals who did not undergo an ultrasound examination revealing positive findings indicative of lateral epicondylitis were excluded, along with other patients as outlined in the Materials and Methods section. These measures were implemented to ensure that the study population was appropriately defined and aligned with the research objectives [18,19]. The sensitivity, positive predictive value (PPV), and false negative of Cozen’s, Mill’s, Maudsley’s, and the selfie tests were analyzed. In this study, it is important to understand the concept of false negatives. False negatives occur when a test incorrectly indicates the absence of a condition when the condition is actually present. With the design of this study, it was not possible to perform an analysis of false positives, given that the required inclusion criterion was a confirmed diagnosis of LE. Sensitivity represents the proportion of true positives, or the percentage of patients who have a positive test result among those who have the disease; in this case, positive ultrasound findings. PPV represents the proportion of true positives among all positive test results, indicating the probability of having the disease given a positive test result. 

## 3. Results

Thirty patients were included in the study, seventeen of who were females (57%). The mean age was 50.1 years old (range of 35 to 68 years). Twenty-seven patients had a dominant right hand, of which sixteen suffered from right elbow pain. In the three left-handed patients, the left elbow was affected. All of the patients reported the onset of their symptoms following casual activity, without any obvious trauma or specific triggers. The average duration of symptoms was 7 ± 3.1 months (range of 2 to 14 months). Patients’ demographics are presented in Table 2.

The mean PRTEE score was 61.5 ± 16.1 (range of 35 to 98) and the mean subjective elbow score was 63 ± 14.2 (range of 30 to 80). Mill’s, Maudsley’s, Cozen’s, and the selfie tests were positive in 26, 25, 29, and 28 patients, respectively. The selfie test has a sensitivity of 0.933, PPV of 0.933, and false negatives of 0.067. 

Table 3 shows the mean PRTEE and subjective elbow scores between patients with positive and negative results for each test. For Mill’s test, 26 patients had a positive result, with a mean PRTEE score of 63.4 and a mean subjective elbow score of 62.1. Among the four patients with a negative Mill’s test result, the mean PRTEE score was lower at 49.5, and the mean subjective elbow score was slightly higher at 68.75. Similarly, for Maudsley’s test, 25 patients had a positive result, with a mean PRTEE score of 64.68 and a mean subjective elbow score of 60.4. Among the five patients with a negative result, the mean PRTEE score was 46, and the mean subjective elbow score was higher at 76. Cozen’s test had the highest number of positive results, with 29 patients, and a mean PRTEE score of 61.8 as well as a mean subjective elbow score of 65. For the single patient with a negative result, the mean PRTEE score was 53 and the mean subjective elbow score was 50. Finally, for the selfie test, 28 patients had a positive result, with a mean PRTEE score of 62 and a mean subjective elbow score of 63.8. Two patients had a negative result, with a mean PRTEE score of 55 and a mean subjective elbow score of 75. The results indicate that among patients with positive provocative tests, the questionnaire scores were higher and the subjective evaluation was lower, indicating a worse level of function and pain. In contrast, patients with negative provocative tests had lower questionnaire scores and higher subjective evaluations, indicating better function and less pain. As the number of patients with negative provocative tests was very small, it was not possible to calculate a *p*-value for the difference in PRTEE and subjective elbow scores between positive and negative tests.

Table 4 presents the diagnostic test performance for all four provocative tests. The sensitivity of each test was measured by the proportion of patients with a positive ultrasound diagnosis who tested positive on a given test. Mill’s test showed a sensitivity of 0.867, meaning that it correctly identified 86.7% of patients with lateral epicondylitis. Maudsley’s test had a sensitivity of 0.833, indicating that it identified 83.3% of cases correctly. Cozen’s test had the highest sensitivity at 0.967, identifying almost all cases (96.7%) correctly. The selfie test had a sensitivity of 0.933, correctly identifying 93.3% of cases. The PPV, or positive predictive value, was calculated by the proportion of patients who tested positive on a given test and had a positive ultrasound diagnosis. The PPV of each test was high, with Cozen’s test showing the highest PPV at 0.97, followed by Mill’s and the selfie tests at 0.87 and 0.93, respectively.

## 4. Discussion

In this study, we aimed to assess the diagnostic accuracy of the selfie test, a novel tool for identifying LE. Mill’s, Maudsley’s, Cozen’s, and the selfie tests were performed, and a majority of patients exhibited positive results. The selfie test demonstrated a high sensitivity of 93% and a positive predictive value (PPV) of 93%. Furthermore, the sensitivity of all tests ranged from 87% to 97%, while their specificity was relatively low, ranging from 3% to 13%. The overall accuracy of the tests ranged from 83% to 93%. These findings suggest that the selfie test holds promise as an accurate diagnostic tool for LE, comparable to other established tests.

LE, which was originally attributed to or affected athletic individuals participating in racket sports, can also affect individuals in their daily activities, which consist of repetitive work with their hands [3]. With the advancement of technology, cell phones have become an integral part of daily life, and usage has steadily increased over the years [20]. Excessive scrolling, clicking, and holding the phone with one hand, often without elbow support, can lead to repetitive movements. This constant use has been linked to increased occurrences of elbow and wrist pain, particularly among younger generations [21,22]. The widespread use of a cell phone may increase the likelihood of clinically manifesting LE while using the phone, especially when taking a selfie; however, it is worth noting that this may simply exacerbate an existing condition or make it more noticeable, rather than being a definitive cause of LE.

The diagnosis of LE often involves the use of passive tests, which are subjective and rely on the skill and experience of the physician conducting the examination. Commonly used provocative tests for diagnosing LE involve applying strain to the common extensor tendon in order to reproduce the patient’s symptoms. These tests include Mill’s, Cozen’s, and Maudsley’s tests [7]. While these tests can be useful tools for diagnosing LE, their accuracy is limited. Not all patients with LE will experience pain during the tests, which can result in false negative results. Conversely, some patients may experience pain during the tests due to other underlying conditions, such as arthritis or nerve entrapment, which can lead to false positive results.

To the best of our knowledge, no prior studies have examined the utility of active tests for diagnosing lateral epicondylitis (LE). Furthermore, there is a lack of studies that have compared the accuracy of active tests with passive tests. The incorporation of active testing in the diagnosis of LE offers potential advantages over provocative and passive tests. By closely simulating a patient’s everyday movements and activities, active testing aids in pinpointing the genuine source of pain without necessitating the presence of an external examiner. This may enhance diagnostic accuracy and provide valuable insights into the condition.

Active testing is a crucial component of telemedicine that refers to the delivery of healthcare services through telecommunication and digital technologies [23]. It allows healthcare providers to remotely assess and diagnose patients by utilizing live video, secure messaging, and other electronic communication tools [14,24]. The increasing prevalence of remote work and virtual interactions has amplified the importance of tools that cater to our evolving daily activities. Sedentary lifestyles, prolonged screen time, and poor ergonomics have resulted in a rise in musculoskeletal disorders and mental health issues. Developing technologies that address these concerns, such as posture-correcting wearables or virtual well-being platforms, can play a pivotal role in promoting physical and mental wellness amidst changing work environments [25].

The benefits of active testing include increased accessibility to healthcare for patients in remote areas, earlier and possibly more accurate diagnoses, resulting in improved outcomes, and increased patient engagement, as patients can take an active role in their own healthcare [13]. Additionally, the development of artificial intelligence (AI) algorithms and machine learning models holds immense potential in analyzing vast amounts of data generated by mobile devices and telemedicine platforms. These technologies can assist healthcare professionals in making accurate diagnoses, predicting disease progression, and personalizing treatment plans based on individual characteristics and lifestyle factors. By leveraging AI in conjunction with mobile health solutions, we can enhance the efficiency and effectiveness of healthcare delivery, ultimately improving patient outcomes [26]. As technology continues to shape our daily activities, it is crucial to adapt our medical tools and tests accordingly. Embracing the integration of mobile devices, telemedicine, and AI-powered solutions can revolutionize healthcare by providing convenient access to medical resources, empowering individuals in their self-care journey and enhancing diagnostic accuracy as well as personalized treatment options.

## 5. Limitations

This study has several limitations. 

In the context of diagnosing lateral epicondylitis (LE), it is important to address the rationale for using ultrasound (US) as the gold standard in this study. Firstly, compared to other orthopedic conditions that may have more definitive and objective diagnostic criteria, there is a lack of a true gold standard for diagnosing LE. This absence of standardization can introduce variability in diagnoses and potentially limit the accuracy of the study’s results. Additionally, it becomes challenging to calculate sensitivity and specificity accurately since there are no true positives or true negatives to serve as reference points.

However, despite the lack of a definitive gold standard, ultrasound has been widely utilized as a reference standard in diagnosing LE due to its advantages. The US provides objective data that can be used for comparison, allowing for more consistent and reliable assessments. It allows for the visualization of affected tendons, highlighting abnormalities such as thickening, calcifications, or tears, which are commonly associated with lateral epicondylitis.

It is worth noting that this particular study encountered a limitation regarding negative diagnoses (negative US findings). As a result, the ability to predict negative values and calculate the specificity of the tests was restricted. This limitation highlights the challenges inherent in studying a condition without a clear gold standard.

It is worth noting that the reliability of US results can be operator-dependent, although we attempted to minimize this influence by exclusively using US performed by an expert radiologist and technician while excluding any externally obtained tests. Another limitation of this study is the small sample size, which limited the statistical analysis and comparison of sensitivity between different tests due to the low number of patients with negative provocative tests. While this study shows promising results regarding the diagnostic performance of the selfie test in the diagnosis of LE, further research is needed to validate these findings.

## 6. Conclusions

The selfie test’s active nature, which allows patients to perform the assessment themselves, could be a valuable addition to the diagnostic process, potentially improving the accuracy of the diagnosis of LE.

## Figures and Tables

**Figure 1 medicina-59-01159-f001:**
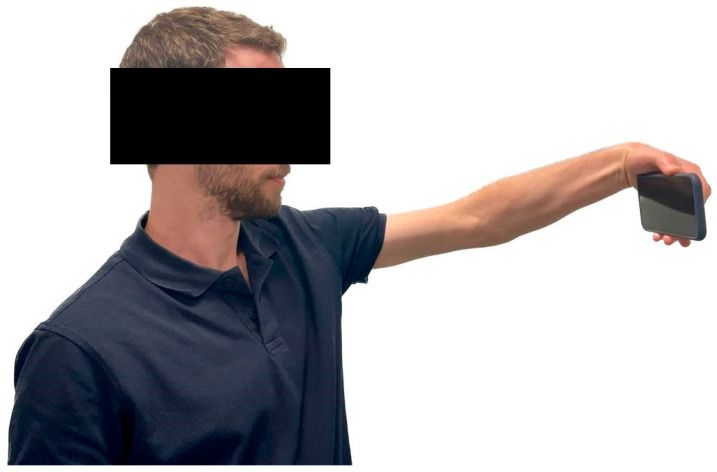
The selfie test: The patient is instructed to hold a cell phone with their elbow fully extended and flex their wrist, independently pressing their thumb on the phone screen or top button. A positive test is indicated by the presence of pain in the lateral aspect of the elbow joint.

**Table 1 medicina-59-01159-t001:** Inclusion and exclusion criteria for study participants.

**Inclusion Criteria**
Patients over 18 years of age
Recurrent or new persistent lateral elbow pain.
Ultrasound findings:
-Thickening of the common extensor tendon at its origin.
-Hypoechoic areas within the tendon indicating degeneration or tears.
-Irregularity of the cortex (bone spurs) of the lateral epicondyle.
**Exclusion Criteria**
Patients under 18 years of age
Diagnosis of cervical radiculopathy or myelopathy.
Neurological disorders.
Associated elbow fractures.

**Table 2 medicina-59-01159-t002:** Patients’ demographics and baseline characteristics.

	*n* = 30
Female (%)	17 (57%)
Male (%)	13 (43%)
Mean age (range)	50.1 (35–68)
Right-handed affected elbow	16 right, 11 left
Left-handed affected elbow	0 right, 3 left
Duration of symptoms (range)	7 ± 3.1 (2–14)
PRTEE score	61.5 ± 16.1
Subjective elbow score	63 ± 14.2

PRTEE; Patient-Rated Tennis Elbow Evaluation.

**Table 3 medicina-59-01159-t003:** Comparison of PRTEE and subjective elbow scores between positive and negative results of provocative tests *.

Test	Positive/Negative	Mean PRTEE Score	Mean Subjective Elbow Score
Mill’s	Positive (26)	63.4	62.1
	Negative (4)	49.5	68.75
Maudsley’s	Positive (25)	64.68	60.4
	Negative (5)	46	76
Cozen’s	Positive (29)	61.8	65
	Negative (1)	53	50
Selfie	Positive (28)	62	63.8
	Negative (2)	55	75

* Due to the very small number of patients with negative provocative tests, it was not possible to calculate a *p*-value for the difference in PRTEE scores between positive and negative tests, and any such calculation may not be reliable.

**Table 4 medicina-59-01159-t004:** Diagnostic test performance for lateral epicondylitis.

Test	Sensitivity
Mill’s	0.867
Maudsley’s	0.833
Cozen’s	0.967
Selfie	0.933

## Data Availability

Data are available upon reasonable request from the corresponding author.

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
