# Peer review of "The “Selfie Test”: A Novel Test for the Diagnosis of Lateral Epicondylitis"

_medicina, 2023, doi:10.3390/medicina59061159_

Round 1

Reviewer 1 Report

1. Image of the persons face should be covered and not identifiable

2. References to be the same format (see number 16 and correct)

3. There must be a description of false positive tests in the statistical analysis

4. All authors should have their ORCID ID's

5. Contributions - specify what each author did and how they contributed

Author Response

  1. Image of the persons face should be covered and not identifiable.

Comment accepted. The figure was edited.

  1. References to be the same format (see number 16 and correct)

Comment accepted. References were edited.

  1. There must be a description of false positive tests in the statistical analysis.

Data regarding false negative and false positive was added to this section and further elaborated.

  1. All authors should have their ORCID ID's

Authors ORCID ID were added and sent to editorial office.

  1. Contributions - specify what each author did and how they contributed.

Author contributions were added to the text.

Reviewer 2 Report

The editors mention to the limitations of the study.

They might improve the quality of the work by suggesting risk mitigations for false positives and negatives. What could be done? they might offer some controls maybe? to eliminate this problem.

The impact of their work, when applied, should be given? what is the benefit? 

What could be improved in the future? etc

Author Response

We appreciate your insightful comments and suggestions regarding potential risk mitigations and improvements for future research. We added and discussed these topics in the revised version of the manuscript.

To address the issue of false positives and negatives in the diagnostic tests for lateral epicondylitis (LE), incorporating controls and additional measures can be valuable. One possible approach is the inclusion of a control group of individuals without LE to compare the test results and evaluate the specificity of the tests. This would help in distinguishing between true positives and false positives. Moreover, implementing standardized protocols and training for healthcare professionals administering the tests can enhance consistency and accuracy.

In terms of the impact and benefits of this work, the use of active testing, such as the Selfie Test, offers several advantages. Firstly, it provides a more realistic simulation of everyday movements and activities, aiding in identifying the actual source of pain and improving diagnostic accuracy. This can lead to earlier detection and treatment, potentially resulting in better patient outcomes. Additionally, incorporating telemedicine and mobile health solutions into healthcare practices can enhance accessibility to medical resources, increase patient engagement, and allow for personalized treatment options based on individual characteristics and lifestyle factors. To further improve in the future, it would be valuable to conduct larger-scale studies with diverse populations to validate the findings and assess the generalizability of the tests. Additionally, exploring the integration of artificial intelligence (AI) algorithms and machine learning models can help automate the analysis of data generated by mobile devices and telemedicine platforms, aiding healthcare professionals in making more accurate diagnoses and personalized treatment plans. In summary, suggestions for risk mitigation, inclusion of controls, and the potential benefits of active testing, telemedicine, and AI-powered solutions have been noted. Continual advancements in these areas and the integration of novel technologies have the potential to enhance diagnostic accuracy, improve patient care, and shape the future of healthcare.